# Isolation and Identification of Biocontrol Bacteria against *Atractylodes Chinensis* Root Rot and Their Effects

**DOI:** 10.3390/microorganisms11102384

**Published:** 2023-09-25

**Authors:** Shouyang Luo, Chunjie Tian, Hengfei Zhang, Zongmu Yao, Zhihui Guan, Yingxin Li, Jianfeng Zhang, Yanyu Song

**Affiliations:** 1Key Laboratory of Mollisols Agroecology, Northeast Institute of Geography and Agroecology, Chinese Academy of Sciences, Changchun 130102, China; luoshouyang@iga.ac.cn (S.L.); tiancj@iga.ac.cn (C.T.); zhanghengfei059868@163.com (H.Z.); yaozongmu@iga.ac.cn (Z.Y.); liyingxin@iga.ac.cn (Y.L.); 2Key Laboratory of Straw Comprehensive Utilization and Black Soil, Conservation College of Life Science, The Ministry of Education, Jilin Agricultural University, Changchun 130118, China; zhangjianfeng06@tsinghua.org.cn

**Keywords:** *Atractylodes chinensis*, root rot, biological control, microbial diversity

## Abstract

Fusarium root rot (FRR) seriously affects the growth and productivity of *A. chinensis*. Therefore, protecting *A. chinensis* from FRR has become an important task, especially for increasing *A. chinensis* production. The purpose of this study was to screen FRR control strains from the *A. chinensis* rhizosphere soil. Eighty-four bacterial strains and seven fungal strains were isolated, and five strains were identified with high inhibitory effects against *Fusarium oxysporum* (FO): *Trichoderma harzianum* (MH), *Bacillus amyloliquefaciens* (CJ5, CJ7, and CJ8), and *Bacillus subtilis* (CJ9). All five strains had high antagonistic effects in vitro. Results showed that MH and CJ5, as biological control agents, had high control potential, with antagonistic rates of 86.01% and 82.78%, respectively. In the pot experiment, the growth levels of roots and stems of *A. chinensis* seedlings treated with MH+CJ were significantly higher than those of control plants. The total nitrogen, total phosphorus, total potassium, indoleacetic acid, and chlorophyll contents in *A. chinensis* leaves were also significantly increased. In the biocontrol test, the combined MH + CJ application significantly decreased the malondialdehyde content in *A. chinensis* roots and significantly increased the polyphenol oxidase, phenylalanine ammonolyase, and peroxidase ability, indicating a high biocontrol effect. In addition, the application of *Bacillus* spp. and *T. harzianum* increased the abundance and diversity of the soil fungal population, improved the soil microbial community structure, and significantly increased the abundance of beneficial strains, such as *Holtermanniella* and *Metarhizium*. The abundance of *Fusarium*, *Volutella*, and other pathogenic strains was significantly reduced, and the biocontrol potential of *A. chinensis* root rot was increased. Thus, *Bacillus* spp. and *T. harzianum* complex bacteria can be considered potential future biocontrol agents for FRR.

## 1. Introduction

*Atractylodes chinensis* is a perennial herb of the Asteraceae family. The dry root of this herb is primarily used as medicine and has the effects of drying dampness to strengthen the spleen, dispel cold symptoms, or improve eyesight. *A. chinensis* is mainly distributed in Asia and Europe [1]. At present, countries with large planting areas of *A. chinensis* include China, India, South Korea, and Thailand [2]. Among them, China is the largest producer of *A. chinensis*, with an area of approximately 200,000 hectares. *A. chinensis* is an authentic medicinal material in Northeast China and the northwestern part of Hebei. Given the unique efficacy of *A. chinensis*, the market demand for this herb has increased dramatically in recent years, but wild *A. chinensis* is in short supply. Therefore, artificial cultivation is urgently needed to supply the market. However, given that cultivation has progressed over the years, disease problems have also become increasingly prominent. Root rot, the most harmful disease in *A. chinensis* cultivation, is mainly caused by *Fusarium oxysporum* and *Fusarium solani* [3]. For *A. chinensis* root rot, the use of chemical agents and physical management does not produce fruitful results but also causes pesticide pollution [4]; eco-friendly strategies for mitigating this devastating disease have not yet been reported. Effective control methods to solve the dilemma of pharmaceutical farmers must be sought [5]. Microbial control has been widely studied because of the high efficiency, low consumption, and environmental friendliness of this method.

Soil microorganisms are mainly responsible for the biological processes that maintain soil health and inhibit plant diseases [6]. In general, after continuous cropping, the soil microflora changes, the microbial community structure becomes unbalanced, beneficial bacteria are inhibited, and harmful bacteria develop in large quantities, resulting in rampant soil-borne diseases [7]. Biological control of plant root diseases can inhibit root pathogens by regulating rhizosphere microflora and adding probiotics [8]. In overcoming plant diseases, *B. subtilis* [9], *T. harzianum* [10], and *B. amyloliquefaciens* [11] play important roles in biocontrol. These species have a control effect on various soil-borne plant diseases; in particular, *Bacillus* and *Trichoderma* have a control effect on different soil-borne plant diseases [12]. However, the prevention and treatment of *A. chinensis* root rot and its growth promotion are seldom reported.

This study intends to screen the abovementioned biocontrol strains from *A. chinensis* rhizosphere soil collected in Tonghua City, China, to test their resistance to *Fusarium oxysporum* (FO). The control effect of *A. chinensis* root rot was initially detected using pot experiments. This study is expected to provide a scientific basis for controlling *A. chinensis* root rot and developing and utilizing disease-resistant probiotics.

## 2. Materials and Methods

### 2.1. Screening of Culturable Biocontrol Strains

Healthy *A. chinensis* rhizosphere soil was randomly collected from Tonghua City (125.20° E, 41.47° N), Jilin Province, China, in October 2021. The diluted plate coating method was used to culture the PDA medium (materials: 200 g potato, 20 g glucose, 18 g AGAR, distilled water at constant volume to 1000 mL; natural pH) and LB medium (materials: 10 g tryptone, 10 g NaCl, 5 g yeast powder, 18 g AGAR, and 1 L distilled water). After assessing the different morphological characteristics, 84 strains of bacteria and seven strains of fungi were isolated. The pure culture of the strain was stored in 50% glycerol (*v*/*v*) at −80 °C for future use.

### 2.2. Assessment of Antagonistic Effect

FO was placed at the center of a cake dish containing the PDA and LB media (http://www.biozj.com, accessed on 30 October 2022), and then the strains were inoculated 2.5 cm away from each other. The other Petri dish was not inoculated with the biocontrol strain; only FO was inoculated as a control. After 12 days of incubation at 28 °C, the length and width of the pathogen colony were measured to evaluate the antibacterial effect of the isolated strains. The inhibition rate was calculated as (D−B)/D (100%) (D: CK, colony diameter; B: Colony width of the treated pathogen). Each treatment was repeated three times. Only the pathogen cake was taken as the control (CK), and the average value of three replicates was taken.

### 2.3. Identification of Strains

For the fungi, 18S rDNA was selected for detection. The paired primer sequences were ITS1 TCCGTAGGTGAACCTGCGG and ITS4 TCCTCCGCTTATTGATATGC. The reaction system (25 µL) was as follows: Taq enzyme (12.5 µL), ITS1 (10 nM at 1 µL), ITS4 (10 nM at 1 µL), DNA template (1 µL), and sterile water (<25 µL). The reaction conditions were as follows: predenaturation at 95 °C for 5 min; denaturation at 94 °C for 30 s and annealing at 55 °C for 30 s. The reaction was extended at 72 °C for 90 s (total: 32 cycles), followed by 7 min at 72 °C.

For the bacteria, 16S rDNA was selected for detection. The paired primer sequences were 27F 50-AGAGTTTGA TCCTGGCTCAG-30 and 1492R 50-T ACGACTT AACCCCAATCGC-30. The reaction system was as follows: 2 × Taq PCR Master Mix (12.5 µL), forward primer and reverse primer (1 µL), and ddH_2_O (9.5 µL). The reaction procedure was as follows: 95 °C for 5 min; 94 °C for 1 min, 55 °C to 58 °C for 1 min, 72 °C for 90 s (total: 30 cycles), followed by 72 °C for 10 min. After PCR amplification, the product was examined by 1.5% agarose gel electrophoresis, and obvious characteristic bands were observed. The amplified product was sent to Shenggong Bioengineering Co., Ltd. (Shanghai, China) for sequencing.

For phylogenetic analysis, the Basic Local Alignment Search Tool (BLAST) algorithm was used to compare the 18S rDNA and 16S rDNA gene sequences with the National Center for Biotechnology Information (NCBI) GenBank entries to obtain homologous sequences. The phylogenetic tree was constructed using the neighbor connection method of MEGA version 7.0.

### 2.4. Identification of Biocontrol Factors and Detection of Indoleacetic Acid Content

The selected strains were inoculated into skim milk AGAR medium, chitin identification medium, and cellulose Congo red medium [13]. Bacterial strains dipped with sterilized toothpicks were connected to the plates of skim milk AGAR medium and chitin test detection medium. After culture at 28 °C for 3 days, a transparent circle was observed, and the d value (transparent circle diameter colony longitude) was calculated. The calculated result was used as the basis for detecting the activity levels of protease, cellulase, and chitin. Here, (++++) is D ≥ 2 cm, (+++) is D = 1–2 cm, (++) is D = 0.5–1 cm, and (+) is D < 0.5 cm.

### 2.5. Evaluation of Growth-Promoting Activity of Strains

The selected strains were inoculated into phosphorus solubilization medium, azotobacter medium, and potassium solubilization medium [14], and their growth trends were observed. After 7 days of culture, the strain presented a growth-promoting effect, and a transparent circle was produced in the screening medium. Then, the effect based on the growth of the bacteria in each Petri dish and the size of the clear circle was determined in terms of whether they could grow in a selective medium (+), whether the clear circle was the same size as the colony (++), whether the clear circle was slightly larger than the colony (+++), and whether the clear circle was more than twice the colony size (++++). Yao’s method was used to determine the indoleacetic acid activity (IAA) of biocontrol bacteria [15].

### 2.6. Pot Experiment for Screening the Biocontrol Potential of Isolated Strains

The soil was sterilized with gamma rays in the pot test. *A. chinensis* seedlings with the same growth and development were planted in the same area and transplanted into pots with three plants per pot, with three pots for each treatment. The eight treatments were as follows: L1 (control), L2 (MH), L3 (FO), L4 (MH+FO), L5 (CJ5 + CJ7 + CJ8 + CJ9), L6 (MH + CJ5 + CJ7 + CJ8 + CJ9), L7 (CJ5 + CJ7 + CJ8 + CJ9 + FO), and L8 (MH + CJ5 + CJ7 + CJ8 + CJ9 + FO). *T. harzianum* suspensions (10^6^ CFU/mL) and Bacillus complex biologic suspensions (2 × 10^7^ CFU/mL) were applied to each pot by irrigation. In addition, the fermentation broth of L7 and L8 biocontrol bacteria was treated with 1:1 (*v*/*v*) compound bactericide. After 60 days of biocontrol colonization, *F. oxysporum* lumps cultured for 5 days were applied. The resistance activity of POD, PAL, PPO, and MDA in *A. chinensis* roots was determined after 8 days.

### 2.7. Promotion Effect of Biocontrol Strains on the Growth of A. chinensis Seedlings

*A. chinensis* seedlings with the same growth and development were planted in the same area and transplanted into pots. Each pot was irrigated with 40 mL 10^7^ CFU/mL biocontrol bacterial suspension. The experiment included three plants per pot and three pots per treatment, totaling three treatments (L2, L5, and L6). Then, the three *A. chinensis* treatments were removed from the pots, and the root thickness and stem length were measured using a ruler. A 1/1000 scale was used to weigh and calculate the dry and fresh weights. The total nitrogen and total phosphorus of plants were measured using a continuous flow analyzer, the total potassium was measured via inductively coupled plasma emission spectroscopy, and chlorophyll was determined via ethanol extraction [16]. The IAA yield of *A. chinensis* leaves was measured according to the method of Yao [15].

### 2.8. DNA Extraction, PCR, and Sequencing

DNA extraction and PCR sequencing were performed on *A. chinensis* rhizospheres soil treated with L3, L4, L7, and L8 to test the effects of inoculated *T. harzianum* and *Bacillus* spp. on the community structure of *A. chinensis* rhizosphere fungi after FO application. Then, the DNA was isolated, and the method of Ji et al. [17] for purification and concentration assessment was adopted. The fungal ITS1 region was amplified using the 5-GGAAGTAAAAGTCGTAACAAGG-3 and 5-GCTGCGTTCTTCA TCGATGC-3 primers. dNTPs (2.5 mmol/L), forward and reverse primers (10 mol/L each), 10 Pyrobest buffer (5 L), Pyrobest DNA polymerase (2 U/L), and dissolved DNA in ddH_2_O (36.7 μL, 1 ng/μL) were included in the kit’s 50 L PCR system. The PCR procedure involved 25 cycles of reaction at 95 °C for 30 s, 56 °C for 30 s, and 72 °C for 40 s over the course of 5 min. Paired-end sequencing was performed on the Illumina-MiSeq platform [18]. The raw data were filtered by deleting junction-contaminated reads, junction-contaminated reads with N-containing reads, and reads with end sequences shorter than 20. (Adjustment of Short reads, v1.2.11). The QIIME2 (From Personal Biology, Changchun, China) classify-sklearn algorithm was used to combine the paired reads obtained by double-end sequencing into a single sequence using the overlap relationship to obtain high-variation region tags. In the overlap region, the minimum match length was 15 bp, and the allowable mismatch rate was 0.1. Reads with no overlap relationship were removed. Then, by utilizing a pretrained basic Bayesian classifier with default parameters, the QIIME2 tool (From Personal Biology, Changchun, China) was used to label the species for each ASV feature sequence. For the species taxonomic annotation of fungal ITS sequences, the UNITE database (Release 8.0, https://unite.ut.ee, accessed on 30 October 2022) was used in conjunction with the Greengenes database (Release 13.8, http://greengenes.secondgenome.com, accessed on 30 October 2022). The confidence level was set to 0.6.

### 2.9. Bioinformatics and Statistical Analysis

In QIIME2, sparsity curves and taxonomic composition were used to detect whether the sample sequencing depth could achieve saturation. The “phyloseq” program for Microbiome Analyst1 was used to export and subsequently calculate the alpha diversity indices, including the Chao1 and Shannon indices. The “VennDiagram” tool in R (v3.6.1) was used to count the shared and unique OTUs among various samples. The changes in microbial structure were identified by principal coordinate analysis (PCoA) based on Bray–Curtis distances. ANOVA was implemented to determine variations in relative abundance at the phylum level. Redundancy analysis was conducted using the “Vegetation” package in R (version 3.6.1) to determine the relationships with microbial communities. A statistical significance test was performed in STAMP analytic software (version 2.1.3) to examine the variations in the relative abundance of fungi across the L3, L4, L7, and L8 treatments. Then, Spearman rank correlation coefficients (|*p*| > 0.6, *p* < 0.05) and correlation coefficients based on total OTUs were adjusted for multiple tests to determine whether changes in microbial communities could be connected to the soil environment. FUNGuild software (Lingbo MicroClass, http://www.cloud.biomicroclass.com, accessed on 30 October 2022) w-as used to investigate the fungal functionalities of diverse samples.

### 2.10. Statistical Analysis

SPSS 24.0 (IBM Corp, SPSS Statistics for Windows, version 24.0. Armonk, NY, USA) was used for ANOVA. Mean values were subjected to Duncan’s multiple range tests at *p* < 0.05. All diagrams were processed and analyzed using Adobe Illustrator CS6 (Adobe Systems Inc., San Francisco, CA, USA) and GraphPad Prism (8.0.2).

## 3. Results

### 3.1. Screening and Identification of Biocontrol Strains

Eighty-four strains of bacteria and seven strains of fungi were isolated from the rhizosphere soil of *A. chinensis*, and the inhibitory effect of FO, a root rot pathogen, was studied. Among them, 13 strains had strong antagonistic activity against FO. In addition, MH, CJ5, CJ7, CJ8, and CJ9 showed significant inhibitory effects (*p* < 0.05) compared with other strains (Figure 1). The highest antagonistic rates of MH and CJ5 against FO were 86.01% and 82.78%, respectively (Table 1).

Morphological analysis and biochemical characterization were performed on the abovementioned five strains with significant antagonistic effects. The colonies of MH cultured on a PDA medium were deep and dense and grew rapidly. After 1–3 days of culture, the mycelium became white and turned green after 4–5 days, and was concentric or ring-shaped; furthermore, the mycelium expanded radially, and the air mycelium was wadded (Figure 2A,B). Microscopic observation of conidium and conidium terrier conidium a globose or ovoid, conidium terrier ampere bottle shaped, base shrinkage, middle), at the top of the finest point, can produce a large number of conidium (Figure 2C,D). The strain was identified as *T. harzianum* by morphological analysis.

The CJ5, CJ7, CJ8, and CJ9 bacteria were Gram-positive (Figure 3). In combination with 18S rDNA, the 16S rDNA underwent sequencing by using the NCBI website (http://www.NCBI.nlm.nih.gov/, accessed on 18 November 2022) and BLAST. The results showed that the 18S rDNA sequence of MH was highly homologous to the sequence of *T. harzianum*, and the homology between CJ5, CJ7, and CJ8 and *B. amyloliquefaciens* was higher than 99% (Figure 4A–C). CJ9 is 99% homologous to *B. subtilis* (Figure 4D). Then, strain-related sequences and their homologous sequences were collected, and phylogenetic trees were constructed according to genetic distance. Phylogenetic analysis classified MH as *T. harzianum* (Figure 2E); CJ5, CJ7, and CJ8 as *B. amyloliquefaciens* and CJ9 as *B. subtilis*.

### 3.2. Biological Control and Growth Promotion Evaluation

As shown in Table 2, all five strains can produce protease, cellulase, and chitinase, among which the CJ5, CJ7, CJ8, and CJ9 four strains have the strongest protease activity. However, MH had strong cellulase and chitin activity. Protease, cellulase, and chitinase are the primary hydrolases of the fungal cell wall. Therefore, *B. subtilis*, *B. amyloliquefaciens*, and *T. harzianum* may have antagonistic effects on pathogenic bacteria by secreting cell wall hydrolase.

As shown in Table 3, all bacteria can solubilize phosphorus, potassium, and nitrogen, and all bacteria can produce IAA, among which MH has the strongest ability to produce IAA, reaching 63.05 µg/mL. IAA is an effective growth hormone for promoting plant growth [19]. Therefore, all five strains of probiotics have the potential for growth promotion (Table 3).

### 3.3. Biocontrol Agent Is Effective in Preventing and Promoting Disease in A. chinensis

Five strains of biocontrol bacteria were used to prepare composite biological agents with different treatments to promote growth in pots. Then, the physiological and biochemical indices of each treatment pot were determined. As shown in Table 4, the fresh and dry weights of *A. chinensis* roots after L6 treatment increased by 78.62% and 58.75%, respectively, compared with the control (Table 4). Meanwhile, compared with the control, the water content in *A. chinensis* roots treated with L2, L5 and L6 increased by 18.64%, 21.20% and 22.14%, respectively. In addition, compared with the control, the fresh leaf and stem weights of *A. chinensis* after the L2 and L6 treatments increased by 62.87% and 85.04%, respectively, and the corresponding stem weights of *A. chinensis* after the L2, L5 and L6 treatments increased by 56.54%, 18.79% and 53.91%, respectively. The leaf and stem water contents in *A. chinensis* increased by 32.87% after the application of complex bactericide L6. The increase in plant water content was more conducive to the absorption of nutrients, and the growth-promoting effects of microbial preparations on the root length, leaf stem, and root diameter of *A. chinensis* were significantly increased. The maximum root length and root diameter of *A. chinensis* after the L2, L5 and L6 treatments significantly increased. Compared with the control, the maximum root length of *A. chinensis* after the L2, L5 and L6 treatments increased by 66.87%, 62.45% and 69.65% and the root diameters increased by 49.72%, 46.76% and 57.43%, respectively; these trends are closely related to the promotion of underground growth. In addition, L2, L5 and L6 significantly increased the aboveground portion length of *A. chinensis* by 37.75%, 30.73% and 57.76%, respectively, which is closely related to the promotion of aboveground growth.

The effects of different microbial preparations on the nitrogen, phosphorus, and potassium contents in *A. chinensis* indicate the abilities of L2 and L6 to effectively increase the total nitrogen levels of plants, which were 11.16% and 21.19% higher than that of the control, respectively. The change in phosphorus content also indicates that L6 can significantly increase the phosphorus content in plant leaves, which is 54.77% higher than that of the control group. Comparisons of the total potassium content in leaves treated with different microbial preparations further indicate that the three treatments of L2, L5, and L6 can significantly affect the increase in potassium, reaching 8.15%, 25.60% and 38.18%, respectively (Table 5).

Then, the IAA and chlorophyll contents were determined to further verify the effects of different microbial agent treatments on *A. chinensis* growth. The IAA contents in *A. chinensis* plants increased significantly after implementing the different microbial preparation treatments. Compared with the control group, the IAA after L2, L5, and L6 treatments increased by 37.72%, 19.67% and 40.85%, respectively. In addition, the application of different treatments of microbial preparations significantly increased the chlorophyll content in *A. chinensis* leaves. Compared with the chlorophyll α in the control, the values increased by 45.44%, 68.27%, and 73.32% after performing L2, L5 and L6 treatments, while chlorophyll β increased by 18.87%, 37.03%, and 44.52%, and the total chlorophyll content increased by 31.74%, 54.40%, and 60.86%, respectively (Table 5).

The results of the data analysis are shown in Table 6. The MDA activity clearly increased after FO infection. Compared with the MDA activity in the control group inoculated with FO, the MDA content after inoculation with L2, L5, and L6 decreased by 33.42%, 13.43%, and 28.68%, respectively. PPO is an osmotic regulatory substance secreted by plants when they are injured or invaded by pathogens, and its activity reflects the ability of the plant to resist disease and defend itself. The experimental data showed that the PPO activity of each treatment group increased significantly after FO inoculation. Treatment with different microbial preparations also increased PPO accumulation in *A. chinensis* roots. Compared with the PPO activity in the control group inoculated with FO, the PPO activity after inoculation with L2, L5 and L6 increased by 24.01%, 21.33%, and 22.53%, respectively (Table 6).

The changes in PAL activity and POD presented trends similar to those of PPO. With FO infection, the PAL and POD activity in all groups significantly increased, and the root defense activity of *A. chinensis* significantly improved, but the degree of increase varied. Compared with the activity of the control group inoculated with FO, the PAL activity increased by 16.24%, 13.62%, and 17.52% after inoculation with L2, L5, and L6, while the POD activity increased by 42.53%, 55.60%, and 55.67%, respectively. MDA, PPO, PAL, and POD were determined using *A. chinensis* leaves inoculated with microbial inoculations. Although the stress resistance index of the plants fluctuated after inoculation with microbial inoculations, the change was not significant compared with that of normally grown *A. chinensis* (Table 6).

### 3.4. Diversity of Microbial Communities

The microbial α diversity in the rhizosphere soil of *A. chinensis* treated with L3, L4, L7, and L8 was determined. Compared with the Chao1 index of L3, the Chao1 indices of L4, L7, and L8 increased, while that of L8 soil significantly increased (Figure 5A). No significant difference was observed in the Shannon index (Figure 5B). The results indicate that the L8 treatment increased the fungal diversity of the *A. chinensis* rhizosphere soil. In addition, the overall patterns of the fungal communities in the rhizosphere soil of *A. chinensis* treated with L3, L4, L7, and L8 were evaluated via the Bray–Curtis distance-based PCoA ranking method (Figure 5C). In the PCoA of fungal communities, the first principal component (PCo1) and the second principal component (PCo2) could explain 25.7% and 15.6% of all variables, respectively. In summary, the fungal composition of the *A. chinensis* rhizosphere soil treated with the L3, L4, L7, and L8 treatments presented varying trends (Figure 5C). The results of further analysis of the changes in the number of ASVs in each treatment in domains, phyla, class, order, family, genus, and species are shown in Figure 5D. After treatment with biocontrol fungi, the number of ASVs in each taxonomy was higher than that of the L3 treatment. Then, a combination of *T. harzianum* and *Bacillus* spp. was applied. The biological control fungi of *A. chinensis* may inhibit the growth of some harmful fungi in the rhizosphere soil to achieve certain control effects.

The changes in the relative abundance of microbial communities at different levels were also evaluated. Then, the response of soil microbial groups to different biocontrol treatments was explored. Figure 6 shows the top 10 phyla in terms of the average relative abundance for each community composition. Ascomycota (80.12% to 80.36%) and Basidiomycota (7.63% to 11.83%) were the most dominant phyla in the four treatments (Figure 6A). In fungal communities, the relative abundance of Ascomycota and Basidiomycota increased in soils treated with L4 and L7 compared with those treated with L3 (Figure 6A). According to the Wayne diagram, the proportions of fungi in soil treated with L4, L7, and L8 were higher than those of unique fungi in soil treated with L3 (Figure 6B–D).

At the fungal genus level, stamp analysis was performed on soil treated with different biocontrol bacteria (*p* < 0.05) to further determine the response of microorganisms to soil treated with different biocontrol bacteria, namely, L3, L4, L7, and L8 (Figure 7). The fungal genus level included 14 different groups of L3 compared with that in L4 (*p* < 0.05); the relative abundance of these 14 different groups was significantly higher than that of L3 (*p* < 0.05). The relative abundances of *Mrakiella*, *Metarhizium*, *Holtermanniella*, and *Cylindrocarpon* in L3 significantly decreased (*p* < 0.01) (Figure 7A). L3 had four different groups compared with L7 (*p* < 0.05). The relative abundances of *Fusarium* and *Volutella* in the soil treated with L7 were significantly decreased, while the relative abundances of *Nectria* and *Ceratobasidium* were significantly increased *(p* < 0.05) (Figure 7B). In addition, compared with the abundance in L3-treated soil, the relative abundances of *Fusarium* and *Helotiales* in L8-treated soil were significantly reduced, whereas the relative abundances of *Pezizella* and *Tilletiopsis* significantly increased (*p* < 0.05) (Figure 7C).

## 4. Discussion

### 4.1. Screening and Identification of Biological Control Strains

Soil-borne diseases are considered to be the primary cause of cropping obstacles, significantly inhibiting plant growth and ultimately reducing crop yield and quality [20]. In recent years, given the good antifungal properties of *B. subtilis*, *B. amyloliquefaciens* [21], and *T. harzianum* [22], these species have been regarded as strains with great application potential in controlling plant fungal diseases. In experiments performed in this research, *B. amyloliquefaciens* (CJ5, CJ7, and CJ8) and *B. subtilis* (CJ9) were identified, and their inhibition rates against FO exceeded 80%. After culturing against FO, the treated soils could compete for the nutritional and spatial sites of FO and inhibit the growth of FO, a situation similar to the results of Nandi and Sen [23]. A fungal strain identified as *T. harzianum* (MH) had a bacteriostatic rate of 86.01% after the plate confrontation assay was performed; when the confrontation culture involved FO, it could rapidly occupy space, absorb nutrients, and occupy the invasion site of pathogens, effectively forming nutritional and spatial competition with pathogens and inhibiting the growth of FO. This situation is similar to the findings of Guzman-Guzman et al. [24].

The growth promotion ability of biocontrol strains identified in this research indicates that biocontrol bacteria produce IAA and dissolve phosphorus, potassium, and nitrogen, which can improve the nutrient absorption of plants. Indoleacetic acid is an effective growth hormone for promoting plant growth, and it exists in *T. harzianum* [25] and *Bacillus* spp. [26]. In the pot experiments, the biocontrol preparations effectively increased the dry weight, fresh weight, plant height, root diameter, and chlorophyll content in *A. chinensis*, improved the biomass, and presented strong growth promotion ability. Cultivations with *T. harzianum*, *B. amyloliquefaciens*, and *B. subtilis* were more helpful to the growth of *A. chinensis*.

*B. subtilis*, *B. amyloliquefaciens* [27], and *T. harzianum* [22] can also produce different antibacterial active substances. Hepatica amylolytica and hepatica subtilis can produce more than ten enzymes, including isubtilin, cellulase, and β-glucanase [28]. *T. harzianum* can produce small molecules of antibacterial peptides and cell wall hydrolases, such as chitinase, cellulase, xylanase, and protease, which can inhibit the growth of pathogenic fungi independently or synergistically with cell wall-degrading enzymes [29]. Protease, cellulase, and chitinase, as biocontrol factors, are compound hydrolases that hydrolyze the cell walls of fungi and inhibit pathogenic bacterial growth, thus achieving antibacterial and disease prevention [30]. In this study, the metabolites of *B. subtilis*, B. amyloliquefaciens, and *T. harzianum* all contained three active components: protease, cellulase, and chitinase. Hence, the three bacteria can produce cell wall hydrolase and other bacteriolytic substances via antagonistic effects, consequently inhibiting the growth of pathogens or directly killing them. Indoleacetic acid is an effective growth hormone for promoting plant growth. In the pot experiments, *B. subtilis*, *B. amyloliquefaciens*, and *T. harzianum* effectively increased the dry weight, fresh weight, plant height, root diameter, and chlorophyll content in *A. chinensis* and increased the biomass; the growth promotion effect was also obvious.

### 4.2. Growth-Promoting Effect of Biocontrol Strains

Nitrogen, phosphorus, and potassium, as the primary synthetic components of proteins and enzymes in plants, play critical roles in plant growth [31]. Nitrogen is also an important component of chlorophyll, which promotes cell division and growth, enhances plant photosynthesis, and promotes stem and leaf growth and fruit development [32]. Therefore, nitrogen content is closely related to chlorophyll content in plant leaves. In the experiment, the nitrogen content and chlorophyll content of *A. chinensis* were significantly increased after L6 treatment. In addition, phosphorus and potassium not only can improve plant root development and increase yield [33] but also have good water transport capacity [34] and plant disease resistance [35,36]. In the experiments, the total phosphorus and total potassium contents in *A. chinensis* significantly increased after L6 treatment, similar to the root length. Furthermore, stem length, root thickness, and water content in *A. chinensis* significantly increased and effectively prevented FO. These results were consistently obtained in this study.

As the primary growth hormone of plants, IAA has an obvious promotion effect on plant cell growth, followed by a strong role in nutrient transport [19]. In addition, some indole derivatives have high broad-spectrum antifungal activity against pathogenic fungi, including *F. graminis*, Sorokinia, Megalosporium, FO, and *Alternaria alternata* [37]. Here, the growth-promoting effects and abilities of L2 and L5 were compared via experiments. L2 promoted *A. chinensis* growth by increasing the IAA content in plants, while L5 promoted the growth of *A. chinensis* by enhancing the photocooperation of *A. chinensis*. The effects of L2 and L6 indicate that the increase in IAA is more conducive to the root growth and stem length of *A. chinensis*, especially since L6 simultaneously increases IAA and chlorophyll in the plant. Among the four treatments, plants treated with L6 had the highest root length, stem length, root thickness, and fresh weight. The stem length of L2 was smaller than that of L6. Furthermore, L6 significantly affected the chlorophyll level of *A. chinensis*. Chlorophyll promotes the growth of *A. chinensis* stems.

### 4.3. Biocontrol Effect of Biocontrol Strains

Membrane lipid peroxidation occurs in plant cells under aging or stress conditions. As one of the primary products of membrane lipid peroxidation, MDA can cross-link lipids, nucleic acids, carbohydrates, and proteins, react strongly with various components in cells, and cause oxidative stress to plant cells to a certain extent, resulting in damage to the cell membrane system [38]. Therefore, MDA activity can be used to determine the lipid peroxidation level in plant cells and the biofilm damage degree to reflect the lipid peroxidation degree in cell membranes and the reaction intensity of plants to adverse conditions; it can also be used as one of the indicators to judge the level of plant injury by pathogenic fungi [39]. Under stress conditions, MDA activity is negatively correlated with plant resistance [40] and positively correlated with cell membrane permeability [41]. As consistently revealed by the experimental results, the MDA activity in *A. chinensis* seedlings significantly increased because of FO infection, and the plant cell membrane suffered a certain degree of oxidative damage. The application of microbial agents effectively reduced the accumulation of MDA, helped alleviate the occurrence of membrane lipid peroxidation in *A. chinensis*, and improved plant stress resistance.

When plants are infected by pathogens, a series of changes usually occur in their physiological, biochemical, and organizational structures, including some active defense reactions related to disease resistance, such as the production of lignin, the synthesis and accumulation of plant protection hormones, the enhancement of resistant enzyme activity, and the production of resistant proteins. PAL, PPO, POD, and other resistant enzymes play important roles [42]. PAL is the most important enzyme in phenolic substances and lignin synthesis [43]. The primary function of PPO is to oxidize phenolic substances to quinones, which are more toxic to pathogens [44]. POD and its isoenzymes play important roles in H_2_O_2_ removal and lignin and phenolic synthesis [45]. Consequently, the activity levels of PAL, PPO, and POD in plant cells have been proposed as physiological indicators of plant disease resistance [42]. In the experiments performed in this research, the PAL and PPO activity in *A. chinensis* seedlings were significantly enhanced after FO infection, and the effects were more significant after the application of microbial preparations. These results indicate that PAL, PPO, and POD played important roles in the inoculation of FO-induced root rot. *A. chinensis* started to be damaged and became infected when the PAL, PPO, and POD activity increased after inoculation with pathogenic bacteria. However, the treatments with microbial preparations slowed the incidence and severity of FO damage in *A. chinensis*, significantly reducing the probability of pathogen infection. These results indicate that the defense enzyme activity of *A. chinensis* seedlings can be enhanced by increasing PAL, PPO, and POD activity while decreasing MDA activity.

### 4.4. Biocontrol Effect of Biocontrol Strains

Soil microbial diversity is often characterized using the Chao1 and Shannon indices. In general, the higher the index is, the richer the microbial species or quantity [46]. In this study, after *Bacillus* spp. and *T. harzianum* were sprayed, the Chao1 and Shannon indices of fungi in the rhizosphere soil of *A. chinensis* were higher than those of the control group with FO alone. This finding indicates that the application of biocontrol bacteria improved the richness and diversity of the *A. chinensis* rhizosphere soil fungal community. The soil fungal community of *A. chinensis* developed in a stable and diversified manner [47], which is consistent with the effect of biocontrol fertilizer on the soil fungal community structure [48].

The soil microbial population and community composition are closely related to the occurrence of plant diseases [49]. On the one hand, the application of biocontrol microorganisms can exert a biocontrol effect via the direct role of the strain itself; on the other hand, it can affect the occurrence of plant diseases by changing the soil microbial community structure [50]. This study found that administering *Bacillus* spp. and *T. harzianum* increased the community abundance of some beneficial microorganisms. At the phylum level, spraying *Bacillus* spp. and *T. harzianum* (L7 and L8) increased the relative abundance ratio of ascomycetes in the soil. Most ascomycetes are saprophytic fungi that play important roles in degrading organic matter [51]. In addition, the application of biocontrol agents can increase the relative abundance of basidiomycetes, while basidiomycetes fungi in soil can promote the decomposition and circulation of soil substances, which are mostly beneficial soil microorganisms [52]. At the genus level, the application of *T. harzianum* (L4) significantly increased the abundance of beneficial fungal genera such as *Holtermanniella* and *Metarhizium*. *Holtermanniella* is a basidiomycete fungus, most of which can biodegrade; this ability is often related to soil pH, organic matter, and other soil nutrients [53]. As one of the important biological insecticides, *Metarhizium* plays an important role in the green control of pests; it not only has a good control effect on underground pests but can also promote the growth and development of crops and improve the resistance of crops to diseases and insects. In addition, some endogenous *Metarhizium* showed antagonistic effects on some plant pathogenic microorganisms [54]. Consistently determined by the experimental results, *Bacillus* spp. and *T. harzianum* (L7 and L8) significantly reduced the abundance of *Fusarium*, *Volutella*, and other pathogenic bacteria. *Fusarium* stimulates the accumulation of phenolic acids in plants, resulting in plant death. Phenolic acids can regulate the soil microbial community, promote the proliferation of pathogenic bacteria, and reduce beneficial bacteria. The experiments also proved that FO is the primary pathogenic bacterium of *A. chinensis*. *Volutella* is less commonly reported domestically, with the root rot of Fusarium spp. Furthermore, mixing accelerates plant decay.

The biocontrol abilities of *Bacillus* spp. and *T. harzianum* to effectively address *A. chinensis* root rot can be attributed to two reasons. On the one hand, biocontrol agents secrete cellulase and chitinase, which destroy the cell wall of pathogenic bacteria and greatly reduce the primary infection source of root rot disease. On the other hand, spraying shield biocontrol agents improves the soil microbial structure, reduces the accumulation of pathogenic bacteria, and increases the abundance of beneficial microorganisms in the soil, thereafter, reducing the infection and harm of pathogenic bacteria.

## 5. Conclusions

Five strains obtained from the rhizosphere soil of *A. chinensis*, namely, *B. subtilis* (CJ9), *B. amyloliquefaciens* (CJ5, CJ7, and CJ8), and *T. harzianum* (MH), were isolated and identified in this research. The antagonistic effects of the five isolates against FO were higher than 80% in vitro, and all of them showed high activity against FO. Then, the growth promotion and disease resistance of the biocontrol bacteria were tested. Four biocontrol bacteria could dissolve phosphorus, potassium, nitrogen, and IAA. *T. harzianum* and *Bacillus* spp. also entailed different biocontrol factors, such as protease, cellulase, and chitinase. In the verification test, the different treatments of *T. harzianum* and Bacillus complex (L2, L5, and L6) not only resisted FO but also positively affected the growth of *A. chinensis*. The total nitrogen, total phosphorus, total potassium, IAA, and chlorophyll contents in leaves increased, subsequently promoting plant growth. In the FO infection test, the application of *T. harzianum* combined with complex *Bacillus* spp. preparations significantly reduced the MDA activity and increased the activity of plant antioxidant enzymes, thereby improving the Fusarium root rot of *A. chinensis*. In addition, the application of *Bacillus* spp. and *T. harzianum* biocontrols increased the abundance and diversity of the soil fungal population in rhizosphere soil, improved the soil microbial community structure, and significantly increased the abundance of beneficial bacteria, such as *Holtermanniella* and *Metarhizium*. The abundance of *Fusarium*, *Volutella*, and other pathogenic bacteria was significantly reduced, and the biocontrol potential of root rot in atractylodes improved. The discovery of these strains offers good prospects for the preparation of inoculants and green planting, as it defines the effects of biocontrol *Bacillus* spp. and *T. harzianum* on soil microbial community structure after application, further providing an important theoretical basis for promoting and applying this biocontrol agent to prevent FRR diseases in large field areas.

## Figures and Tables

**Figure 1 microorganisms-11-02384-f001:**
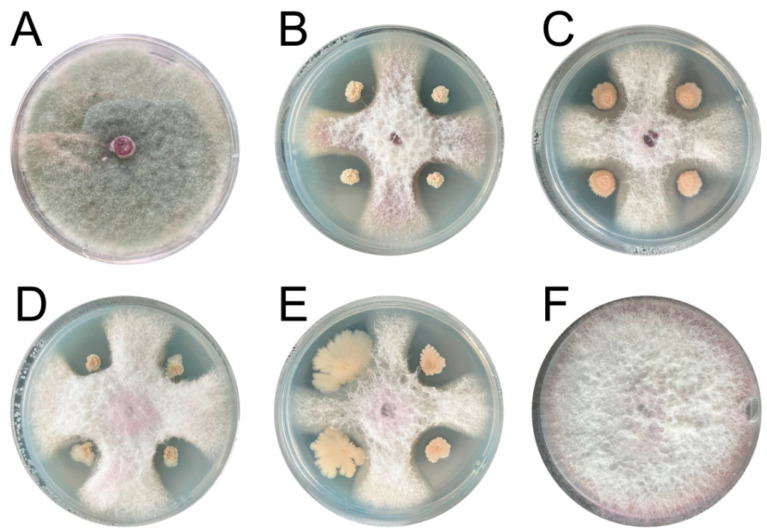
Strains with significant antagonistic effect. (**A**) is the effect diagram of MH against FO; (**B**) is the effect diagram of antagonism between CJ5 and FO; (**C**) is the effect diagram of antagonism between CJ7 and FO; (**D**) is the effect diagram of antagonism between CJ8 and FO; (**E**) is the effect diagram of antagonism between CJ9 and FO; (**F**) stands for FO contrast.

**Figure 2 microorganisms-11-02384-f002:**
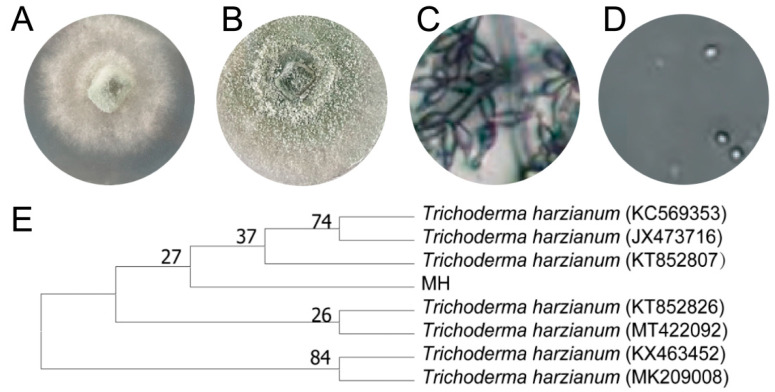
Screening and identification of *T. harzianum*. (**A**) only on PDA (initial stage); (**B**) Colony reverse on PDA (late stage); (**C**) Conidiophore; (**D**) Conidium; (**E**) hylogenetic tree of MH.Where (**C**) is 40× for microscope observation and (**D**) is 100× for microscope observation.

**Figure 3 microorganisms-11-02384-f003:**
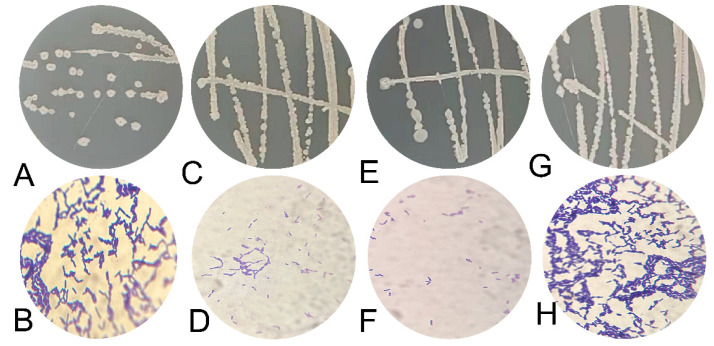
The morphological characteristics of the four isolated bacteria were observed with 100× oil mirror. Individual bacterial forms and grade staining of strain CJ5 in (**A**,**B**), respectively; individual colonies and grade staining of strain CJ7 in (**C**,**D**), respectively; individual colonies and grade staining of strain CJ8 in (**E**,**F**), respectively; individual colonies and grade staining of strain CJ9 in (**G**,**H**), respectively (n = 3).

**Figure 4 microorganisms-11-02384-f004:**
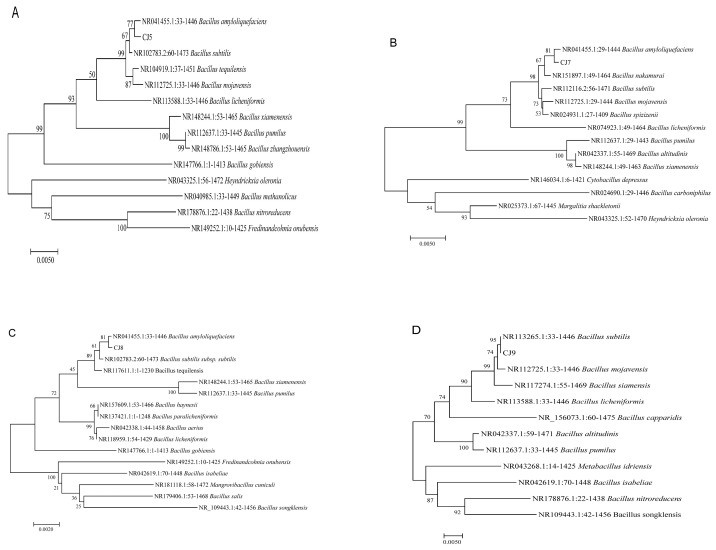
Phylogenetic tree of four biocontrol bacteria. (**A**) CJ5 is *B. amylolyticus*; (**B**) CJ7 is *B. amylolyticus*; (**C**) CJ8 is *B. amylolyticus*; (**D**) CJ9 is *B. subtilis*.

**Figure 5 microorganisms-11-02384-f005:**
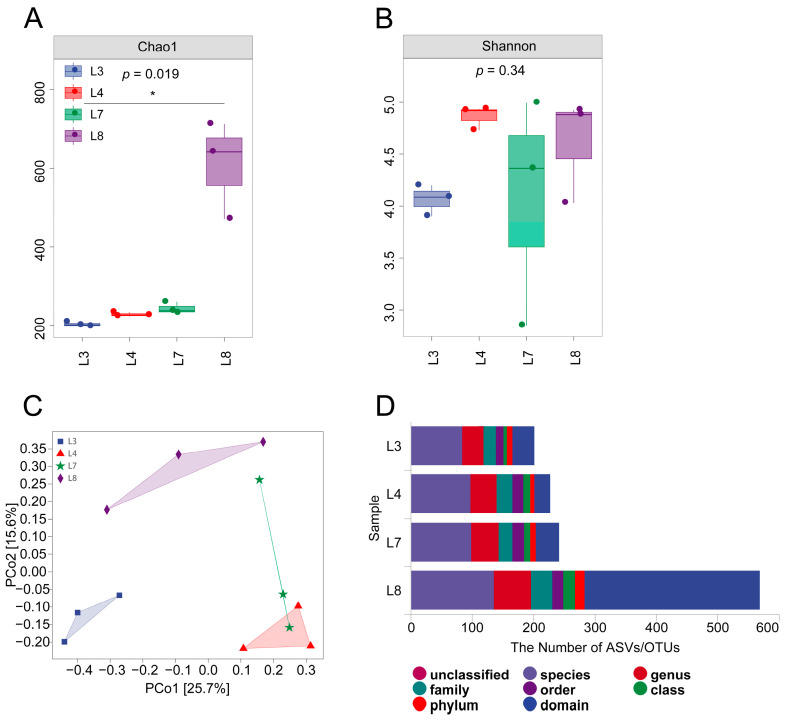
Chao1 and Shannon indices of fungal (**A**,**B**) communities. * above the bar indicates a significant difference between the samples. The differences between the sample groups were significant by one-way ANOVA, followed by Duncan’s multirange test, *p* < 0.05 principal coordinate analysis (PCoA) of fungal (**C**) communities. PCoA was analyzed according to the weighted UniFrac distance of the fungal community. The percentage in the brackets represents the percentage of the sample variance data that can be explained by the corresponding axis. Each dot represents a sample, and different colored dots indicate different groupings. (**D**) represents the number of fungal ASVs at each taxonomic level under four treatments. L3 means FO, L4 means MH + FO, L7 means CJ5 + CJ7 + CJ8 + CJ9 + FO, and L8 means CJ5 + CJ7 + CJ8 + CJ9 + MH + FO (n = 3).

**Figure 6 microorganisms-11-02384-f006:**
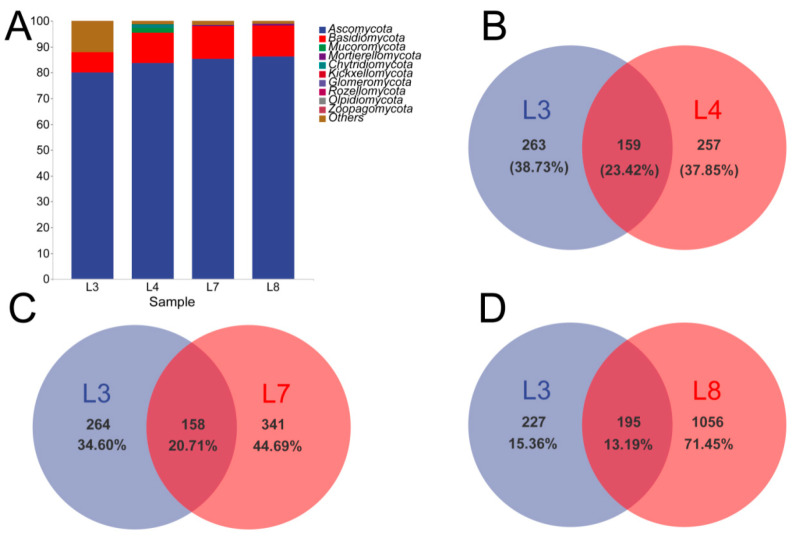
Gate level of the top 10 fungi (**A**) in average relative abundance, abscess is the name of each sample, ordinate is the average relative abundance of each dominant group, Venn diagram of fungi (**B**–**D**) from different organisms treating ASV in soil. L3 means FO, L4 means MH + FO, L7 means CJ5 + CJ7 + CJ8 + CJ9 + FO, and L8 means CJ5 + CJ7 + CJ8 + CJ9 + MH + FO (n = 3).

**Figure 7 microorganisms-11-02384-f007:**
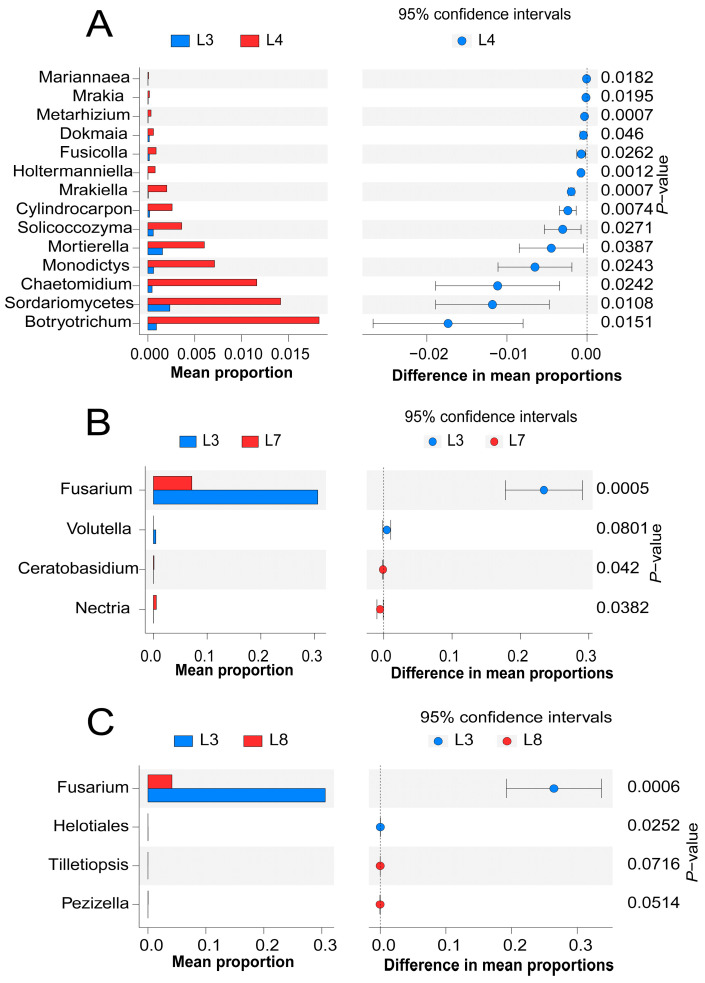
Stamp analysis of different treatment sample difference groups (95% confidence intervals, *p* < 0.05). (**A**) represents the significant response between L3 and L4, (**B**) represents the significant response between L3 and L7, and (**C**) represents the significant response between L3 and L8.L3 means FO, L4 means MH + FO, L7 means CJ5 + CJ7 + CJ8 + CJ9 + FO, and L8 means CJ5 + CJ7 + CJ8 + CJ9 + MH + FO (n = 3).

**Table 1 microorganisms-11-02384-t001:** Antagonistic rate of strains against FO plate screened from rhizosphere soil of *A. chinensis*. The values are means and standard error (n = 3).

Name	Antagonism Rate (%)	Error Value
MH	86.01 a	0.48
CJ1	76.31 cde	1.13
CJ2	74.16 e	0.96
CJ3	67.41 f	0.96
CJ4	75.55 de	0.90
CJ5	82.78 b	0.96
CJ6	76.64 cd	1.76
CJ7	82.77 b	1.16
CJ8	80.07 b	0.64
CJ9	81.40 b	1.38
CJ11	75.21 de	1.76
CJ14	75.21 e	1.02
CJ17	76.01 de	0.70

Different lowercase letters showed significant differences among treatments (*p* < 0.05).

**Table 2 microorganisms-11-02384-t002:** Biocontrol factor capacity of strains.

Factor	Name	Diameter of Transparent Ring (cm)	Error Value (cm)	Diameter of the Colony (cm)	Error Value (cm)	D Value	Activity Level
Protease activity	MH	4.20	0.05	2.60	0.03	1.60	+++
	CJ5	5.40	0.07	3.00	0.06	2.40	++++
	CJ7	5.00	0.10	1.00	0.12	4.00	++++
	CJ8	4.90	0.07	1.00	0.08	3.90	++++
	CJ9	5.00	0.08	1.40	0.03	3.60	++++
Cellulase activity	MH	3.20	0.05	3.10	0.03	0.10	+
Chitin activity	MH	2.40	0.06	0.90	0.03	1.5	+++
	CJ5	1.00	0.03	0.20	0.05	0.8	++
	CJ7	0.80	0.05	0.20	0.04	0.6	++
	CJ8	1.30	0.02	0.30	0.03	1.0	++
	CJ9	0.80	0.05	0.20	0.02	0.6	++

“++++”: D ≥ 2 cm; “+++”: D = 1–2 cm; “++”: D = 0.5–1 cm; “+”: D < 0.5 cm (n = 3).

**Table 3 microorganisms-11-02384-t003:** The growth promoting ability of strain.

Strain Name	Potassium Releasing	Solubilizing Phosphorus	Nitrogen Fixation	IAA Capability (µg/g)
MH	-	-	-	63.05
CJ5	+	+	+	16.66
CJ7	+	+	+	27.56
CJ8	+	+	+	17.39
CJ9	+	+	+	14.68

“+” means having this ability, “-” means not having this ability (n = 3).

**Table 4 microorganisms-11-02384-t004:** Growth promotion effect of biocontrol bacteria on underground parts of *A. chinensis*. L1 indicates a blank control without any treatment, L2 indicates MH treatment alone, L5 indicates the addition of the compound preparation of CJ5, CJ6, CJ7, and CJ9, and L6 indicates the addition of the compound preparation of MH and CJ5, CJ6, CJ7, and CJ9. The values are means ± standard error (n = 3).

Treatment	L1	L2	L5	L6
Root fresh weight (g)	2.91 ± 0.54 b	5.16 ± 1.33 b	5.26 ± 1.68 b	13.60 ± 1.35 a
Root dry weight (g)	1.37 ± 0.54 b	1.36 ± 0.30 b	1.29 ± 0.21 b	3.31 ± 0.82 a
Root water content (%)	53.54 ± 12.19 b	72.18 ± 9.58 a	74.74 ± 4.29 a	75.68 ± 5.45 a
Stem fresh weight (g)	0.50 ± 0.01 c	1.55 ± 0.75 b	0.82 ± 0.17 bc	3.39 ± 0.35 a
Stem dry weight (g)	0.25 ± 0.01 c	0.65 ± 0.28 a	0.29 ± 0.05 b	0.58 ± 0.16 ab
Stem water content (%)	49.66 ± 1.52 b	57.33 ± 3.35 b	63.08 ± 12.64 b	82.53 ± 6.58 a
Longest root length (cm)	2.87 ± 0.31 c	15.00 ± 0.60 b	13.73 ± 0.64 b	13.40 ± 0.53 a
Stem length (cm)	9.47 ± 0.45 c	15.50 ± 2.43 b	13.67 ± 0.58 b	22.50 ± 2.18 a
Root width (cm)	1.07 ± 0.15 b	2.13 ± 0.23 a	2.03 ± 0.35 a	2.63 ± 0.76 a

Different lowercase letters showed significant difference among treatments (*p* < 0.05).

**Table 5 microorganisms-11-02384-t005:** Growth promotion effect of biocontrol bacteria on aboveground parts of *A. chinensis*. L1 indicates a blank control without any treatment, L2 indicates MH treatment alone, L5 indicates the addition of the compound preparation of CJ5, CJ6, CJ7, and CJ9, and L6 indicates the addition of the compound preparation of MH and CJ5, CJ6, CJ7, and CJ9. The values are means ± standard error (n = 3).

Treatment	L1	L2	L5	L6
TN (mg/kg)	15,050.29 ± 674.53 c	16,943.18 ± 105.68 b	16,700.45 ± 321.54 bc	19,134.34 ± 708.24 a
TP (mg/kg)	1685.80 ± 116.54 b	1723.73 ± 408.13 b	2071.62 ± 57.74 b	3726.47 ± 238.80 a
TK (mg/kg)	7903.85 ± 57.74 c	8618.81 ± 419.21 b	10,631.81 ± 393.70 b	12,989.92 ± 2125.36 a
IAA (μg/g)	40.80 ± 1.39 c	65.64 ± 2.52 a	50.82 ± 2.33 b	69.00 ± 1.00 a
Chlorophyll α (mg/g)	4.93 ± 0.18 d	9.04 ± 0.18 c	15.55 ± 0.89 b	18.54 ± 1.68 a
Chlorophyll β (mg/g)	7.80 ± 0.09 d	9.63 ± 0.46 c	12.41 ± 0.52 b	14.11 ± 0.98 a

Different lowercase letters showed significant difference among treatments (*p* < 0.05).

**Table 6 microorganisms-11-02384-t006:** Resistance of biocontrol bacteria to *A. chinensis*. L1 means blank control without any treatment, L2 means MH treatment alone, L3 means FO treatment alone, L4 means MH + FO treatment, L5 means the addition of CJ5 + CJ7 + CJ8 + CJ9 composite bacterial preparation treatment, L6 means the addition of MH + CJ5 + CJ7 + CJ8 + CJ9 composite preparation treatment, L7 means CJ5 + CJ7 + CJ8 + CJ9 composite bacterial agent + FO, L8 represents the effects of MH + CJ5 + CJ7 + CJ8 + CJ9 compound preparation + FO on MDA, PPO, PAL and POD of *A. chinensis*, respectively. The values are means ± standard error (n = 3).

Treatment	MDA	PPO	PAL	POD
L1	5.55 ± 0.39 d	22.33 ± 1.68 c	13.34 ± 0.42 c	13.34 ± 0.42 d
L2	5.59 ± 1.00 d	22.73 ± 1.33 c	12.20 ± 1.59 c	12.20 ± 1.59 d
L3	10.81 ± 1.02 a	35.27 ± 2.20 b	15.56 ± 0.26 b	15.56 ± 0.26 c
L4	7.16 ± 0.43 c	46.47 ± 2.42 a	18.58 ± 0.35 a	18.58 ± 0.35 b
L5	5.50 ± 0.39 d	27.47 ± 4.56 c	13.21 ± 1.55 c	13.21 ± 1.55 d
L6	5.39 ± 0.86 d	20.60 ± 1.44 c	12.74 ± 0.68 c	12.74 ± 0.68 d
L7	9.31 ± 0.36 b	45.40 ± 5.10 a	18.04 ± 0.62 a	18.04 ± 0.62 a
L8	7.67 ± 0.22 c	45.73 ± 2.72 a	18.91 ± 0.98 a	18.91 ± 0.98 a

Different lowercase letters showed significant differences among treatments (*p* < 0.05).

## Data Availability

Not applicable.

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
