# Peer review of "Isolation and Identification of Biocontrol Bacteria against Atractylodes Chinensis Root Rot and Their Effects"

_microorganisms, 2023, doi:10.3390/microorganisms11102384_

Round 1

Reviewer 1 Report

The manuscript entitled " Isolation and identification of biocontrol bacteria against Atractylodes chinensis root rot and their effects” is appropriate for the journal. It is an original and relevant contribution that traces the methodology of bioprospecting of local microbial control agents, to promote their use in the control of phytopathogens. I consider a very complete characterization study, too much work in the project, which gives scientific solidity to the research. Most of the comments are formatted, and are specified in the manuscript (pdf)

Very good work for the authors, congratulations.

The manuscript can be accepted for publication, with minor corrections.

The manuscript requires minor editorial corrections in the English language

Author Response

请参阅附件。

Reviewer 2 Report

The paper entitled “Isolation and identification of biocontrol bacteria against Atractylodes chinensis root rot and their effects” describes the efficacy of different bacterial and fungal strains in the biocontrol of FRR. Although the point is interesting, the paper needs major revision before being acceptable for publication in microorganisms for the following reasons:

1-    The authors check the efficacy of bacterial and fungal strains against FRR, therefore, the title should be revised to highlight the contents.

2-    The abbreviations should be mentioned completely the first time, (for instance, see lines 14, 75

3-    In lines 2, 14, and 138, the scientific names must be italic, please check and revise throughout the manuscript.

4-    Lines 27 -31, is these results or conclusion, please check and revise.

5-    Line 32, “Thus, Bacillus spp. and T. harzianum complex bacteria”, T. harzianum is the fungal strain.

6-    The authors should be revising the hypothesis and aim of the study to be clearer than presented at the end of the introduction section.

7-    The introduction section needs improvement to highlight the efficacy of rhizobacterial strains against plant diseases. The following references are helpful: https://doi.org/10.1515/bmc-2021-0020; https://doi.org/10.1515/bmc-2021-0019.

8-    The units and abbreviation should be standardized (for instance, lines 76 and 77, 1000 mL, 1 L, it is the same).

9-    Line 87, what is the meaning of CK?

10- Subsection “2.2. Assessment of antagonistic effect” should be rewritten scientifically.

11- Line 92, the ITS primers mentioned for fungal identification are for ITS sequence analysis, not for 18S rDNA analysis, please check and revised.

12-   Subsection “2.4. Identification of biocontrol factors and detection of indoleacetic acid content”, is for enzymatic activity only, therefore this title should be corrected. Also, where is the indoleacetic acid detection?

13- Subsections “2.4” and 2.5” should be merged under “plant growth promoting activity.”

14- The components of media used should be mentioned.

15- Please add the accession numbers for identification organisms.

16- The resolution of Figure 4 should be improved. Also, bacterial identification should be merged on phylogenetic trees as well as fungal strains.

17- The title of Table 2 should be rephrased as an enzymatic activity to highlight its content.

18- In Table 2, why cellulase activity was measured for MH only?

19- In Table 3, please clarify the concentration of tryptophan added to broth media to detect the IAA concentration.

20- The conclusion should be concise to highlight the main conclusions and future prospective studies.

moderate editing in English language is needed

Reviewer 3 Report

See atteched file

Reviewer 4 Report

The article "Isolation and identification of biocontrol bacteria against Atractylodes chinensis root rot and their effects" describes the screening of biocontrol strains from A. chinensis rhizosphere soil collected in Tonghua City, China. In order to increase the usefulness and significance of the study, it needs a major revision before being considered suitable for readers and there are some points to overcome for acceptance.

The title of the paper is too ambiguous to understand what the authors are trying to convey. It is recommended to change title. (E.g. Isolation and identification of biocontrol bacteria against Fusarium oxysporum root rot and their effects).

In this article, the author focused on isolation and identification of biocontrol strains from A. chinensis rhizosphere soil, to test their resistance to Fusarium oxysporum. However, the introduction of these sections is quite plain, not in deep. It is recommended to tone up the introduction section.

Use consistent units (line 76 and 77). Use either 1000 mL or 1L.

Recommended to use consistent style for °C (Check throughout the manuscript).

Along with pot experiment it is essential to perform the analytical methods for optimizing the culture conditions. It is recommended to perform the RSM experimental design (PB design), Steepest ascent design and central composite design essential to establish optimal pH, temperature, and rotary shaker speed requirements for growth of the bacterial isolates.

It is recommended to improve the image quality of figure 4.

Please check the alignment of table 2, 4 and 5.

It is suggested a moderate English revision by an English native speaker in order to polish text from typos and imperfections.

Unwanted spacing and typo mistakes throughout the manuscript. Need to be check and correct carefully. (E.g. Line 66).

It is suggested a moderate English revision by an English native speaker in order to polish text from typos and imperfections.

Round 2

Reviewer 2 Report

The authors carefully answered all issues, and the manuscript is suitable for publication in its current form.

Moderate editing of English language required

Reviewer 4 Report

The authors have addressed all the concerns raised in the previous version of the manuscript and the quality has much improved after incorporating required modifications. Therefore, the manuscript may be considered for publication in this Journal.